# Complete Lung Ultrasound Using Liquid Filling: A Review of Methods Regarding Sonographic Findings and Clinical Relevance

**Frank Wolfram * and Thomas G. Lesser**

Department of Thoracic and Vascular Surgery, SRH Wald-Klinikum Gera, Teaching Hospital of Friedrich-Schiller University of Jena, 07548 Gera, Germany; Thomas.Lesser@srh.de

\* Correspondence: Frank.Wolfram@SRH.de



**Featured Application: Lung sonography and sonographic guided pulmonary interventions using partial lung liquid filling.**

**Abstract:** (200w) Lung ultrasound (LUS) is widely used for the diagnosis of pulmonary diseases such as solid nodules and consolidations in contact with the pleural cavity. However, sonography for processes of central disease remains impaired due to total sound reflection at the air tissue interfaces in the ventilated lung. These acoustic barriers can be overcome by replacing intra-alveolar air with liquid. Such filling has been reported using perfluorocarbon, saline or emulsions out of those. In order to achieve acoustic access enabling the use of LUS, complete gas free content is required. Such lung tissue - liquid compound will have untypical physical properties that might impact upon the sonographic visualization of central structures. Up to now, the filling of the lung has been reported for very specific applications and not classified regarding their sonographic findings. This work was therefore motivated to review the literature for methods of lung liquid instillation, classifying their methodological strength and limitations for achieving acoustic access and sonographic findings. Finally, their use for ultrasound based clinical applications will be discussed and the need for research will be outlined.

**Keywords:** lung ultrasound; One Lung Flooding; perfluorocarbon

---

## 1. Motivation

Ventilated lung parenchyma is well known as a total acoustic absorber [1] that impairs intra-pulmonary sonography of consolidations, lung tumors, and central lymph nodes, vessels, and bronchi. Despite these limitations, lung ultrasound (LUS) is widely used in clinical praxis for the exclusion of pneumothoraxes [2], monitoring of extravascular water content [3], or diagnosis of malignancies in contact with pleural cavity [4]. Sonography has shown a high precision comparable to CT for detection and characterization of lung nodules in contact with chest wall [5]. These applications reflect acoustic changes of the pleural cavity and their direct adjacent lung parenchyma. Approaches use shear wave elastography showed feasibility of nodule detection in pleural proximity [6]. However, their sonographic morphology and location to sensitive structures such as vascularity remains unknown.

Unlike with other organs, the gas content of lung parenchyma can be manipulated and therefore acoustic access is generated. Atelectatic lung can achieve sonographic access, but the resulting random collapse is associated with loss of anatomic orientation.

In contrast, instillation of liquids keeping the lung expanded overcomes this problem. Such liquid administration using PerFluoroCarbon (PFC) [7] or saline [8] are used for diagnostic or therapeutic

purposes in pneumology for decades. The fundamentals of lung filling for the use of LUS regarding their sonographic findings have never been summarized and classified. These liquids have different physical and physiological properties, which impacts upon ultrasound imaging and clinical applicability. Therefore, this work was motivated to review the literature regarding lung filling methods and the sonographic findings. Furthermore, the future impact on clinical applications and need for further research will be discussed.

## 2. Perfluorocarbon (PFC) Filling

### 2.1. Methodology

The instillation of lung using PFC based liquids was introduced early in pneumology and neonatal care. Herein, their specific physical-chemical properties are used as an inert volatile liquid, denser than water with low surface tension as well as high oxygen/carbon dioxide solubility. Instilled hydrophobic and biocompatible PFC is not resorbed by perfusion or tissue and remains in the alveolar airspace. Under continued ventilation, PFCs transports oxygen and carbon dioxide, and therefore supports maintaining the gas exchange.

Pulmonary PFC administration is clinically established for the management of acute lung distress syndrome (ARDS) of preterm infants [9] as an effective technique improving the gas exchange with less barotrauma in comparison to gas ventilation.

Recent work showed optimization and standardization strategies to control exogenously the pulmonary gas exchange and temperature [10]. The so called liquid lung ventilation (LLV) is performed partially on an isolated lung wing (PLV) or can be applied to both lungs, which is called total lung ventilation (TLV). Such lung ventilation manages the entire oxygenation and gas exchange, requiring exogenously PFC processing and ventilation control (Figure 1). Despite its use for treatment of ARDS, the PFC filled lung can serve as a heat exchanger. This can lead to applications in oncology as intrapulmonary hyperthermia [11] or as hypothermia for the management of acute traumatic conditions [12].

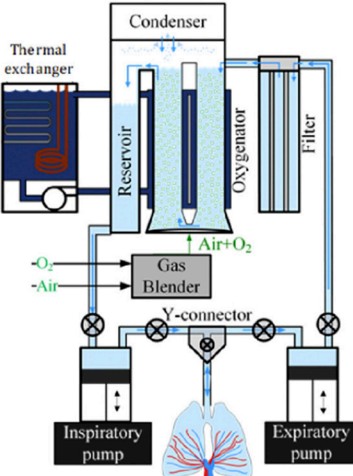

**Figure 1.** Schematic structure of a perfluorocarbon (PFC) liquid lung ventilator, requiring gas exchange of $O_2$ and $CO_2$ using condenser, ventilation of liquids to lung by separated pups for inspiration and expiration, and temperature control. Adapted with permission from Kohlhauer et al. [10].

The use of diagnostic imaging in PFC filled lung was hypothesized for Magnet Resonance MRI [13] or Computer Tomography (CT) [14] imaging. The applicability of pulmonary PFC administration for ultrasound imaging could be assumed due to the gas solubility, absorbing alveolar air and the safety of PFC administration even as an intravenous contrast agent [15].

*2.2. Sonographic Imaging*

The use of PFC lung filling has been intensively investigated and applied in pneumology for decades, but only a few authors considered its further use for sonographic imaging. Initially, Sekins et al. [16] hypothesized the use of sonography for the guidance of interventions in a patent. Later work investigated the acoustic properties of PFC liquids (see Table 1) for such sonographic use in large animal sheep models [17]. Herein, a strong hyperechoic character of PFC filled lung, incapable of visualizing lung structure was reported. Only trans-pulmonary imaging of the beating heart, in proximity to the chest wall, was described.

Later work investigated PFC filling in vivo by Lesser et al. [18] using porcine models and Degnan et al. [19] in infants. In these undertakings, visualization of central structures was not found and sonography appear as white lung syndrome with low penetrability in absence of A Lines (Figure 2). Table 2 summarizes the literature with corresponding sonographic finding of a PFC filled lung.

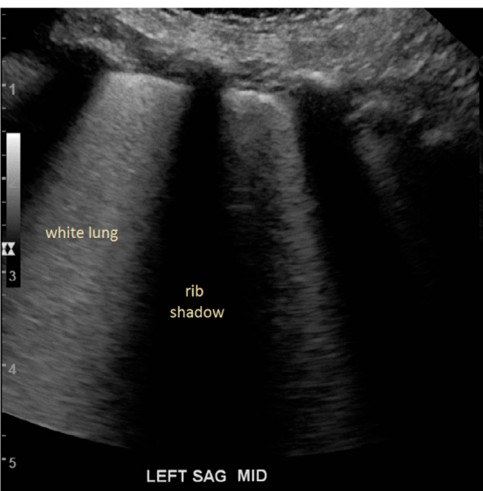

**Figure 2.** Sonographic image of the lung during static PFC administration in infants showing white lung between the rib shadows without the visualization of lung structure and absence of A-lines. (With permission from Degnan et al. [19]).

**Table 1.** Summary of published values for speed of sound, impedance as well as attenuation ($A f^1$) for several PFC liquids, saline @35 °C as well as alveolar tissue. The reflection coefficient of intensities is based on the impedance mismatch to alveolar lung tissue.

| Liquid Used | Speed of Sound [m/s] | Impedance [MRayl] | Reflection Coefficient | Attenuation A [dB/cmMHz] | N |
|---|---|---|---|---|---|
| PFC - Perflubron [17] | 580 | 1.08 | 4.7% (0.42 dB) | 0.22 | 1.8 |
| PFC- FC 77 [21] | 560 | 0.96 | 7.9% (0.75 dB) | 0.35 | 1.6 |
| PFC-Decalin [17] | 620 | 1.25 | 1.0% (0.087 dB) | 0.20 | 1.3 |
| Perfluorohexane [21] | 482 | 0.79 | 13.5% (1.26 dB) | 0.35 | 1.6 |
| Saline (35 °C) [23] | 1520 | 1.53 | 0.03% (0.026 dB) | 0.0014 | 2.0 |
| Alveolar tissue [22] | 1640 | 1.72 | 0% | 0.21 | 1.1 |

**Table 2.** Summary of literature related to sonographic imaging of lung during liquid instillation.

| Paper | Year | Method/Model/Liquid | Findings |
|---|---|---|---|
| Sekins et al. | 2004 | TLV—sheep PFC—Perflubron | - trans pulmonary cardiac visualization<br>- maximum penetration depth 2–4 cm<br>- bright echogenicity of lung/white lung<br>- no bronchial and vascularity detected |
| Degnan et al. | 2018 | PLV—Infants PF—Perflubron | - bright echogenicity of lung/white lung<br>- no bronchial and vascularity detected |
| Lesser et al. | 1998 | OLF—porcine model PFC—PF77 | - bright echogenicity of lung/white lung<br>- no bronchial and vascularity detected |
| Lesser et al. | 1999 | OLF—porcine model Saline 0.9% | - unimpaired lung ultrasound<br>- visualization of simulated nodules |
| Lesser at al. | 2013 | OLF—porcine model and ex vivo human lung saline 0.9% | - complete lung imaging<br>- hypoechoic lung tumors<br>- demarcation of adjacent organs<br>- hypoechoic bronchial and vasculature |
| DePew et al. | 2014 | OLF—human cadaver saline 0.9% | - limited lung imaging<br>- residual gas<br>- sub-centimeter imaging of a forceps |
| Lesser et al. | 2018 | OLF—porcine model and ex vivo human lung saline 0.9% | - unimpaired lung imaging<br>- visualization of simulated nodules<br>- hypoechoic lung tumors<br>- bright echogenicity of puncture needle |

## 2.3. Clinical Relevance

Due to the impaired sonography, specific clinical indications for the use of LUS during PFC liquid ventilation have not been investigated. Despite these limitations, the use of diffuse hyperthermia insonated from cylindrical or piston transducer of low frequency (500 kHz–1 MHz) have been investigated in PFC liquid lung without sonographic guidance [17].

## 2.4. Discussion

Lung filling with PFC is safe, with a low complication rate, and represents an accepted modality that experienced pneumological centers are used to handling. Limiting long recovery time due to their biological inertness requires complete vaporization of intrapulmonary liquid [20]. Sonography of PFC filled lung has been inadequately studied thus far. The underlying assumption of complete gas exchange has not been proven.

It is important to notice that PFC has a much lower speed of sound than solid tissue (Table 1). Images based on time of flight (TOF) assumption would be falsified and require adoption for image reconstruction. Such a correction algorithm would need to distinguish normal tissue with typical water-like speed of sound and PFC filled lung to derive realistic images representing anatomic

dimensions. In addition, the reported visualization depth of 2–4 cm in PFC filled sheep lung [17] was estimated using such general sonography. Therefore, a realistic penetration depth of only 1–3 cm can be assumed.

All authors reported a bright echogenic character of PFC lung, similar to white lung syndrome. The sonographic impairment cannot be explained by PFC attenuation, which is in range of 0.3–0.4 dB/cm/MHz [17,21], slightly lower than solid tissue (0.5–0.8 dB/cm/MHz). When imaging liquid PFC the medium appears hypoechoic similar to water with clear visualization of backstop structures [19]. This indicates multiple reflections of ultrasound waves due to the impedance mismatch in flooded lung as a PFC-tissue compound. Given an impedance of 1.7 MRayl and an attenuation of 0.2 dB/cm MHz of atelectatic lung [22], the reflection coefficients for ultrasound intensities at a PFC interface varies between 1%–13% (Table 1). Theoretically, multiple reflections of ultrasound waves would lead to a 10–130 dB intensity loss within 1 cm of lung penetration (Ø100 μm alveoli) and therefore impair sonographic imaging. In addition, the speed of sound difference between lung parenchyma and PFC would lead to refraction and focal distortion for using any focused ultrasound application (HIFU).

These facts highlight the unsuitability of PFC filled lung out for sonographic imaging. Despite the acoustic high intensity loss, an image reconstruction algorithm would be required for realistic imaging correcting of TOF differences between liquid and tissue. PFC-Decalin might provide the best properties for sonographic visualization but haven't as yet been used for ultrasound imaging in the lung so far.

## 3. Saline-Based One Lung Filling (OLF)

### 3.1. Methodology

Instillation of the lung with physiological saline is used for the removal of pathologic catabolite products [24]. However, such lung lavage does not necessarily generate acoustic access in a quality required for sonographic imaging. In contrast to PFC, saline does not absorb gas and therefore filling can only be performed in one lung requiring the ventilation of the contralateral side (Figure 3). Such OLF (One Lung Filling) requires lung separation and one lung ventilation (OLV) using double lumen tubes (DLT) as commonly performed in thoracic surgery.

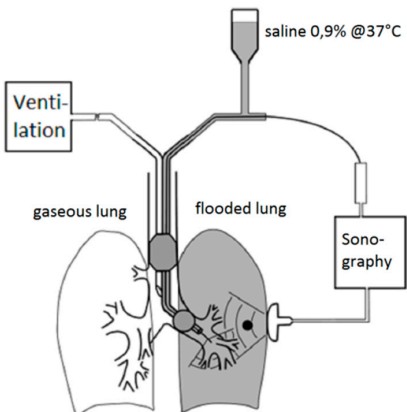

**Figure 3.** One lung flooding (OLF) scheme where one lung wing is filled with saline, while the ventilated contralateral lung is connected to a ventilation system with transcutaneous/endobronchial sonographic imaging. (Adapted with permission from Lesser et al. [25]).

Acoustic impedance of saline is almost identical to tissue with low attenuation, avoiding impedance mismatch and scattering of ultrasound waves when passing the flooded lung parenchyma (Table 1). Initially, Lesser et al. [25] used physiological saline for these reasons resulting in an unimpaired sonographic visualization of central pulmonary structures in a porcine model. The work was intended for sonographic guiding during thoracoscopic or video-thoracoscopic surgery (VATS) where atelectasis was induced by a chest incision followed by the filling of the lung [26]. Motivated by these results,

non-invasive strategies proofing OLF without chest incision was investigated. A slower lung filling time of 20 min was documented, but imaging results were unimpaired similar to filling after atelectasis.

OLF is reversible and the liquid is drained out partially followed by immediate re-ventilation using the same settings for booth lungs. The remaining saline is being absorbed completely by perfusion within approximately 30 min [27]. The safety of OLF was well investigated and seen as not causing deficiencies of oxygenation and pulmonary haemodynamic [28]. In addition, inflammatory response [29] and surfactant loss [30] were in the physiological range.

### 3.2. Imaging Findings

During stable flooding in-vivo with saline, unimpaired imaging without remaining air was achieved. The lung parenchyma appears homogeneous with enhanced echogenicity, where vessels and bronchi are well demarked as hypoechoic structures [18]. Trans-pulmonary imaging visualized adjacent organs such as the liver or heart (Figure 4). Doppler imaging in flooded lung is feasible and useful to detect vessels (Figure 5). No limitations regarding penetration depth were documented. High frequency linear probes (10–15 MHz) were recommended for high resolution imaging such as for small bronchi or vessels [22] (Figure 6). Using ex-vivo models and simulated intra-pulmonary nodules in-vivo, a reliable nodule detection of primary lung cancer as well as metastases were found. The tumor mass appears hypoechoic, surrounded by hyperechoic flooded lung, resulting in a high detection rate of about 100% of malignant pulmonary nodules (Figure 7), as well as simulated nodules [31]. Interestingly, the visualization is aggravated only for the bronchoalveolar cell carcinoma. The specific growth along the alveolar surface caused an echo-enhanced appearance with similar characteristics to flooded lung parenchyma [18].

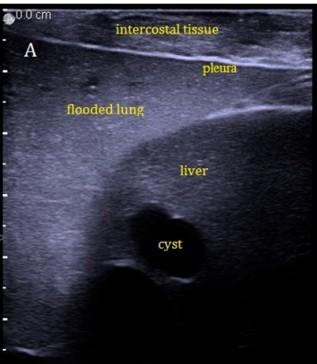

**Figure 4.** Sonographic imaging during OLF in-vivo with clear visualization of hyperechoic lung, well demarked from hypoechoic liver.

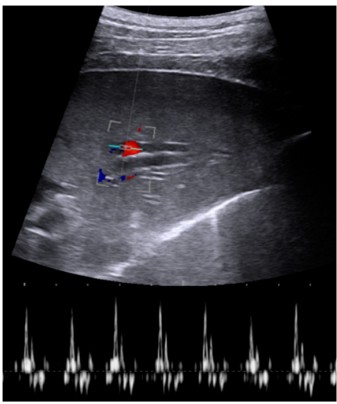

**Figure 5.** Doppler imaging of central through flooded lung. Pulmonary artery show typical flow characteristics and adjacent bronchus with absence of Doppler signal.

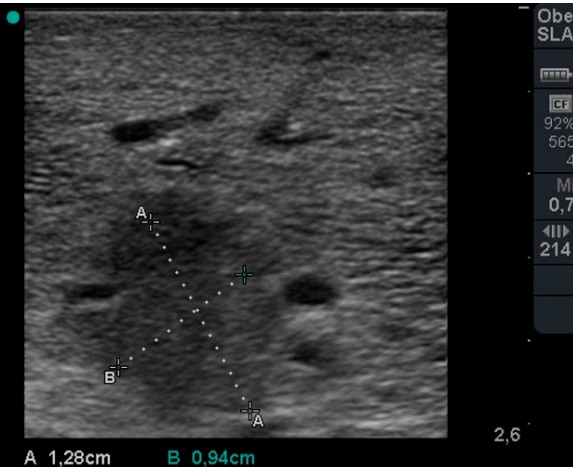

**Figure 6.** Sonographic imaging of hypoechoic Non-Small-Cell Lung Carcinoma (NSCLC) in saline flooded lung using ex-vivo human lung models (With permission from Lesser et al. [18]).

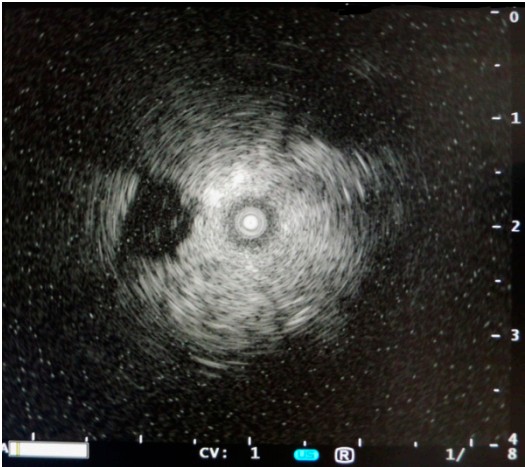

**Figure 7.** Endobronchial sonographic imaging of intrapulmonary nodal metastasis in saline flooded lung using ex-vivo human lung models.

### 3.3. Clinical Relevance

So far, OLF in-vivo was been performed only using large animal models. The promising imaging results have not brought OLF into clinical practice. Therefore, consecutive work investigated minimal invasive procedures, which justified OLF. Herein, sonographic guided tissue sampling was successfully evaluated, resulting in a high (98%) puncture rate [32]. Complications such as pneumothoraxes, which occurs frequently during lung puncture in a ventilated condition, was not found during OLF. The liquid's gravity causes expansion of the lung and avoids pneumothoraxes. These results indicate the favorability of applicator-based procedures during OLF in comparison to ventilated lung. Therefore, the future use of such interstitial based ablation (e.g., MW, RFA, LITT, Electrophoresis) using ultrasound guidance should be conducted in future research.

Ideally, thermal ablation should be performed non-invasively. Substantial work investigating the acoustic conditions in saline flooded lung revealed ideal conditions for non-invasive ablation using therapeutic ultrasound (HIFU). Saline flooded lung provides a low attenuation, minimizing intensity loss and increasing HIFU temperature induction in targeted central cancer tissue. It was found that therapeutic ultrasound intensities interact with malignant tissue inducing lethal thermal damage. The same acoustic dose targeted to the healthy parenchyma causes a non-lethal temperature rise,

therefore leaving the healthy lung parenchyma unaffected [22]. Even acoustic cavitation not causes erosive damage to lung parenchyma at intensities below than 10.000 W/cm$^2$ [33].

In addition to HIFU ablation of pulmonary malignancies, Xie et al. [34] proposed the use for OLF and HIFU for the treatment of tuberculosis. Using the lung as a trans-pulmonary path, targets in adjacent organs such as in the upper liver dome could be targeted by therapeutic ultrasound [35]. Recent work investigated optimal HIFU settings and providing treatment planning modalities [36]. Challenging, are optimal design settings, avoiding beam defocusing and rib heating, which requires modern HIFU modalities such as non-thermal histotripsy [37] or endobronchial insonation [38].

### 3.4. Discussion

It was found that saline-based OLF results in unimpaired visualization of lung in the absence of remaining gas content using B-Mode sonography. The limited visualization during OLF using cadavers indicates the importance of perfusion for gas resorption [39]. Saline as an acoustic matched liquid to parenchyma is an ideal medium where effects of refraction and scattering as well as attenuation of ultrasound waves are minimized in lung as a liquid-tissue compound. OLF has been investigated consecutively, starting with physiological and safety aspects via acoustic modelling to clinical applications. It required two decades to pave the way path from the idea towards evaluation of its clinical use for diagnostic and therapeutic ultrasound applications.

The high sonographic nodule detection rate and clear demarcation from vascular and bronchial structure is valuable, not requiring contrast agents or fusion techniques. During OLF, anatomic orientation is given due to visualization of adjacent organs. For complementation of the sonographic findings, modern imaging modalities such as elastography or contrast agents should be investigated in future work, improving visualization of bronchoalveolar cell carcinoma.

It is likely that OLF will not be applicable for a diagnosis requiring full sedation and insertion of a DLT. Its use for sonographic guidance of therapeutic interventions might justify the invasivity. Favorable for interventions during OLF, are the high nodules detection rate, with the demarcation of vascularity and reduction of ventilation associated motion [40]. The reduction of intrapulmonary perfusion in the flooded lung [28] reduces the risk of hemorrhage during puncture and eliminates possible perfusion associated heat flux when performing thermal ablation. In combination with real-time sonography providing high resolution, low costs, and absence of radiation, LUS guiding during OLF is more favorable than CT or MRI.

Most promising for the clinical use of OLF is the combination of diagnostic lung ultrasound for guidance of therapeutic ultrasound ablation (HIFU) [41]. From such a non-invasive ablation of central lung malignances the field of pulmonary oncology might benefit from. However, the translation to human subjects is hindered due to the unavailability of clinical HIFU systems capable of trans-costal focusing. Several authors have proposed methods to focus HIFU behind the rib cage [42], but these methods have not been implemented in clinical systems to date.

A limitation of OLF, arises from the procedure duration. This operating window was investigated so far only up to 60–90 min, which is sufficient for sonographic guided tissue sampling, MWA or catheter-based ablation (typical 10–15min). During HIFU ablation the procedure time might exceed these limits depending on ablation speed and tumor size. Future work should investigate methodological optimization to expand OLF duration without the impairment of acoustic access.

## 4. Summary and Outlook

Lung ultrasound has been used for decades, becoming a valid clinical tool for characterizing peripheral-pleural lung consolidations. In contrast to CXR, which determines density variations, LUS can differentiate atelectasis, pneumonia, lung infarct, and is useful for the diagnosis of heart failure and pneumothorax.

In order to achieve unimpaired ultrasound imaging of the entire lung, a replacement of gas by a liquid fraction is needed. So far, two methods have been investigated for the purpose of endobronchial

liquid instillation. Herein, OLF using saline generates unimpaired sonographic imaging. The use of inert PFC has physiological advances although it is acoustically inappropriate. The liquid must have identical properties, with regard to speed of sound and impedance, as the alveolar tissue. Such properties so far can only be provided by physiological saline (see Table 1). Using PFC liquids with deviating acoustic properties, sound diffraction and scattering appear in the flooded alveolar parenchyma, which impairs any sonographic imaging. Therefore, no progress has been seen in the improvement of PFC based ultrasound applications in lung over processes performed with saline based OLF. There are no further needs in optimizing the acoustic conditions under saline OLF. Sonographic tumor detection rate, as well as visualization of intra-parenchymal vascular and bronchial structures are superior, and imaging in depth is slightly better than in abdominal organs.

Given the valuable B Mode visualization during saline OLF, most of the accepted diagnostic and sonographic guided interventions are applicable in the lung as practiced for abdominal organs such as the liver. Particular for minimal invasive interventions, OLF is justifiable and beneficial. Performing OLF was initially motivated out of pulmonary oncology for the management of central lung nodules using sonographic guidance during VATS. Later work showed the feasibility of complete new minimal-invasive treatment strategies, such as non-invasive HIFU ablation of central lung nodules.

However, despite the promising results, OLF related research was based on pre-clinical models. Therefore, studies showing the advances of OLF based sonographic guided interventions in a clinical setting are of high demand. The investigation of puncture hit- and complication rate of sonographic guided biopsy during OLF in comparison with CT would be the logical next step. Herein, all the techniques are clinically established, and the outcome is immediately measurable.

**Author Contributions:** Conceptualization, F.W. and T.G.L.; Literature Research, F.W. and T.G.L.; Writing, F.W.; Reviewing-Editing, F.W. and T.G.L. All authors have read and agreed to the published version of the manuscript.

**Funding:** There were no funding sources for this work.

**Conflicts of Interest:** The authors declare no conflict of interest.

## Abbreviations

| | |
|---|---|
| $CO_2$ | Carbone Dioxide |
| CXR | Chest X Ray |
| CT | Computer Tomography |
| DLT | Double Lumen tube |
| FUS | Focused Ultrasound Surgery |
| HIFU | High Intensity Focused Ultrasound |
| LUS | Lung Ultrasound |
| MRI | Magnet Resonance Imaging |
| OLF | One Lung Flooding |
| $O_2$ | Oxygen |
| PLV | Partial Lung Liquid Ventilation |
| PFC | PerFluoroCarbon liquids |
| TLV | Total Lung Liquid Ventilation |
| VATS | Video Assisted Thoracoscopic Surgery |

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
