# Peer review of "Complete Lung Ultrasound Using Liquid Filling: A Review of Methods Regarding Sonographic Findings and Clinical Relevance"

_applsci, doi:10.3390/app10020574_

Round 1
Reviewer 1 Report
This paper was therefore motivated to review the literature for methods of lung liquid instillation, classifying their methodological strength and limitations for achieving acoustic access and sonographic findings. The major part of the research key points (LLV, PLV, and TLV) as well as lung sounds are well mentioned and reviewed in this paper. The paper is readable, I don't feel any doubts about this paper. However, the progress and evolution of various methods can be described in more detail, rather than limit the comparison of many methods. And put some critical things about the author's forward-thinking and suggestions.
Author Response
To Reviewer 1
Thank you very much for the evaluation of our manuscript. Please find enclosed a revised version of the manuscript, which has been modified as recommended, together with a point-by-point response to the issues raised by the reviewers. We appreciate the very constructive comments, which significantly helped to improve the manuscript. In addition, the grammar was corrected by a native English.
However, the progress and evolution of various methods can be described in more detail, rather than limit the comparison of many methods.
Response: We partly agree. There are just two methods found so far to replace air with liquid medium in lung for the purpose of achieving ultrasound access (PFC, saline OLF). Booth methods have been used by several authors to investigate sonographic findings and to derive clinical applications. The manuscript is therefore structured in two parts outlining the motivation, findings and clinical relevance for each of those two strategies separately. In the final discussion 4.0, a direct comparison is added. Booth methodologies are performed from different groups in different times, completely independent, therefore no binding source of progress between those can be documented.
In each part at the methodology and discussion section, the historical evolution is outlined, which is more evident for OLF (see line 136-146) than for PFC (line 90-93) lung filling. Here the standard way of PFC liquid administration was used as in pneumology. Due to the poor imaging findings during PFC filling, no attempts were made by the authors to progress- evolve the procedure for imaging purpose. However, we added recent literature to demonstrate improved PFC lung filling schemes and methodology even though the authors intended pulmonary applications not sonographic use. (see line 58-59, [10]). The fact regarding progress and evolution is more detailed explained (see line 256-59).
And put some critical things about the author's forward-thinking and suggestions.
Response: We agree. Some critics have been addressed in each discussion section showing limitations. These aspects are more detailed formulated (see line 101-103; 116-119; 217-18; 225-230). In addition the new section 4, also summarizes the main critics with respect to PFC (see line 252-255) and OLF applications and points to the necessary clinical investigations (see line 264-78).

Reviewer 2 Report
The manuscript „Complete Lung Ultrasound using liquid filling: a review of methods regarding sonographic findings and clinical use” by Wolfram et al., reviews the two different approaches for imaging the whole lung with ultrasound used so far. The manuscript starts with an overarching motivation and introduction and then describes and discusses each method in a different section. It is well written and easy to read.
Overall, this is a nicely written manuscript, which gives a good overview on the two potential method of lung ultrasound imaging. In my opinion, it would be beneficial to have a stronger emphasis on the general approach of ultrasound imaging of the lung. Why would one want to do this? Which applications would benefit from it? What are general approaches to this? Which properties does a fluid to flood the lung ideally have? Most of the questions you answer throughout the manuscript, but the information is scattered in different places. In addition to this, I would recommend to have a general discussion on lung ultrasound imaging in general before the conclusion section (which should be chapter 4 instead of section 3.5!). This discussion could also give some hints in which direction further research could be performed to enable ultrasound imaging in lungs.
Per-Fluoro-Carbons have been used as ultrasound contrast agents since decades (see e.g. Satterfield et al, Comparion of different PFCs as ultrasound contrast agents, Invest Radiolog 1993). It might be helpful to also summarize some of the findings from back then, since it might be an explanation for most of the “features” of ultrasound imaging in PFC-filled lung.
Minor:
Line 53-4: this sentence is grammatically incorrect. Replace transporting by transports or correct otherwise.
Line 74: what do you mean by patent draft? It looks like this patent is published, thus, this is not a draft anymore.
Figure 2: it might be helpful to label some of the structures in the image. I assume, the shadowing is arising from ribs. If so, please mention this in the caption and text.
Line 154: What does high frequency mean in the context of lung imaging? Could you provide approximate values for the frequency?
Line 189-90: please rephrase this sentence, since it is difficult to understand the meaning of it.
Author Response
To Reviewer 2
Thank you very much for the evaluation of our manuscript. Please find enclosed a revised version of the manuscript, which has been modified as recommended, together with a point-by-point response to the issues raised by the reviewers. We appreciate the very constructive comments, which significantly helped to improve the manuscript. In addition, the grammar was corrected by a native English.
Overall, this is a nicely written manuscript, which gives a good overview on the two potential method of lung ultrasound imaging. In my opinion, it would be beneficial to have a stronger emphasis on the general approach of ultrasound imaging of the lung.
Why would one want to do this? Which applications would benefit from it? What are general approaches to this? Which properties does a fluid to flood the lung ideally have?
Most of the questions you answer throughout the manuscript, but the information is scattered in different places. In addition to this, I would recommend to have a general discussion on lung ultrasound imaging in general before the conclusion section (which should be chapter 4 instead of section 3.5!).
Response: We agree. Chapter 3.5 is misleading as it concludes in summary booth flooding and imaging approaches (saline; PFC). A section 4 summary and conclusion is added instead of section 3.5. Herein the requested motivation and emphasis (Why, Which, what, optimal liquid properties) are consecutively formulated. (see line 240-270)
This discussion could also give some hints in which direction further research could be performed to enable ultrasound imaging in lungs.
Response: We agree. In each chapter a summary is included that points out the need for further research. Such as for PFC based filling using optimal liquids with more ideal acoustic properties (see Line 118-119) as well as for saline OLF regarding extension of flooding time (see line 227-30), use of elastography (line 215) and applicator based interventions (line 185-187).
In addition to request of reviewer 1, the central need for future research showing how to implement the sonographic interventional procedures based on OLF into the clinical practice is addressed in section 4 (line 264-68).
Per-Fluoro-Carbons have been used as ultrasound contrast agents since decades (see e.g. Satterfield et al, Comparion of different PFCs as ultrasound contrast agents, Invest Radiolog 1993). It might be helpful to also summarize some of the findings from back then, since it might be an explanation for most of the “features” of ultrasound imaging in PFC-filled lung.
Response: we partly agree. The mentioned literature uses PFC as intravenous injected contrast agents, which not impacts the composition of lung. The main aspect of this manuscript serves the gas replacement in lung by a liquid for sonographic purposes (Edition of Lung Ultrasound). The imaging findings of intravascular PFC in sonographic accessible organs such as liver are not comparable to a complete lung filling. I addition, Satterfield et al showed an increased echogenity of liver during PFC contrast agent administration, while PFC filled lung is completely inaccessible. We are concerned that this comparison (intravenous vs intrapulmonary PFC administration) might confuse the reader! Therefore Satterfield et al is cited to demonstrate the physiological inertness of PFC and applicability in humans and started the historical evaluation of PFC use also for lung. (See Line 66- 68)
Minor:
Line 53-4: this sentence is grammatically incorrect. Replace transporting by transports or correct otherwise.
Response: done see line 53
Line 74: what do you mean by patent draft? It looks like this patent is published, thus, this is not a draft anymore.
Our response: The patent is filed and drafted but not granted. This indicated the patent status A1. Often patents drafts are not validated regarding their novelty or failed in that process. Then the draft remain published. The status (A1) in the patent number clarifies this issue. In order to avoid confusion we refer to the citation in the patent database independent of its status. See line 75 (deleted -draft)
Figure 2: it might be helpful to label some of the structures in the image. I assume, the shadowing is arising from ribs. If so, please mention this in the caption and text.
Response: done see Figure 2 and caption
Line 154: What does high frequency mean in the context of lung imaging? Could you provide approximate values for the frequency?
Response: Agreed. The frequency range of the used linear transducer of 10-15MHz is added. See line 155
Line 189-90: please rephrase this sentence, since it is difficult to understand the meaning of it.
Response: Agreed: The important message from the investigation of HIFU interaction in lung was that the thermal dose is induced in the solid cancerous tissue only. The same therapeutic ultrasound intensity applied to the flooded lung parenchyma causes a non-lethal temperature rise and therefor the HIFU interacts only with malignant tissue not with the lung parenchyma itself. The sentence is re formulated. See line 190-92.
